# A Robot Floating Grinding and Rust Removal Approach Based on Composite Force-Position Fuzzy Control

**DOI:** 10.3390/s25072204

**Published:** 2025-03-31

**Authors:** Tao Li, Qun Sun, Chong Wang, Xiuhua Yuan, Kai Wang

**Affiliations:** School of Mechanical and Automotive Engineering, Liaocheng Univesity, Liaocheng 252000, China; 2320230106@stu.lcu.edu.cn (T.L.); sunqun@lcu.edu.cn (Q.S.); yuanxiuhua@lcu.edu.cn (X.Y.); 2021404952@stu.lcu.edu.cn (K.W.)

**Keywords:** robot grinding, floating end-effector, rust removal, fuzzy differential prior PID, composite force-position control

## Abstract

The removal of rust from large equipment such as trains and ship hulls poses a significant challenge. Traditional methods, such as chemical cleaning, flame rust removal, and laser rust removal, suffer from drawbacks such as high energy consumption, operational complexity, and poor mobility. Sandblasting and high-pressure water jet rust removal face issues such as high consumable costs and environmental pollution. Existing robotic grinding systems often rely on precise measurement of the workpiece surface geometry to perform deburring and polishing tasks; however, they lack the sufficient adaptability and robustness required for rust removal operations. To address these limitations, this study proposes a floating grinding actuator scheme based on compound force-position fuzzy control. By implementing simplified path-point planning, continuous grinding and rust removal can be achieved without requiring the pre-measurement of workpiece geometry data. This solution integrates force and laser displacement sensors to provide real-time compensation for path deviations and ensures adaptability to complex surfaces. A fuzzy derivative-leading PID algorithm was employed to control the grinding force, enabling adaptive force regulation and enhancing the control precision. Rust removal test results demonstrate that under varying advancing speeds, fuzzy derivative-leading PID control can significantly reduce fluctuations in both the grinding force and average error compared to traditional PID control. At a speed of 40 mm/s, excellent control performance was maintained, achieving a rust removal rate of 99.73%. This solution provides an efficient, environmentally friendly, and high-precision automated approach to rust removal using large-scale equipment.

## 1. Introduction

In industrial production, large-scale equipment, such as ship hulls, bridge construction materials, chemical pipelines, and large mechanical equipment casings, are exposed to complex environments for a long time. Surface corrosion seriously affects performance and service life. Grinding robots can conduct regular maintenance rust removal on these devices, precisely removing rust layers while avoiding damage to the normal structure and critical components of the equipment, thereby reducing the downtime and maintenance costs caused by equipment failures. Traditional manual grinding and rust removal methods are inefficient and involve high labor intensity. Chemical rust removal, flame rust removal, and laser rust removal are only suitable for small-sized and complex-shaped parts because of their high energy consumption and inconvenience in operation and control [1,2,3]. Sand blasting for rust removal and high-pressure water jet rust removal both have significant disadvantages, such as high costs and the difficulty of handling waste sand and wastewater [4,5]. In the continuous rust removal of large-scale workpieces, robots offer distinct advantages through precise path planning and accurate force control, ensuring uniform grinding and rust removal quality while being environmentally friendly and free of significant consumables, thus demonstrating greater promotional value.

As the advantages of robotic grinding and rust removal are becoming increasingly obvious, in-depth research has been conducted in many countries. Pushcorp (Richardson, TX, USA), SHL (Hamburg, Germany), and MEPSA (Barcelona, Spain) have developed grinding robot end-effectors, which can be mounted on the end of the robot and can complete the grinding and rust removal process [6,7,8,9]. Philobody Robotics developed an active haptic flange [10] that calculates the displacement using a control algorithm and is compensated by a pneumatic actuator to ensure stable rust removal and grinding force. Mohammad et al. [11] designed a force-controlled end servo mechanism that detects the contact force in real time through a force sensor, which can be effectively applied to robotic grinding and rust removal operations and can reduce the end vibration and ensure the quality of rust removal. Wang [12] and others proposed a friction-driven pipe outer-wall grinding robot, in which the spinning and rotating motions of each grinding tool are driven by the same motor, which achieves full coverage of the outer pipe surface. Koh [13] used a contact feedback polishing tool for grinding and rust removal operations on overhead pipelines to address utility pipeline rusting.

During the grinding and rust removal operation, force control accuracy and stability are crucial, and advanced control strategies can precisely regulate the end grinding force; therefore, research on control strategies has also attracted attention. Haiqing C. [14] designed an active flexible force control end-effector with first-order differential force feedforward control, which reduces the force control error by 70% compared with the traditional rigid end-effector and can be used for robotic grinding and rust removal of curved parts. Chosei R. [15] proposed an optimal parameter-finding algorithm based on SAC (Soft-Actor-Critic) to enable the robot to stably perform constant-force grinding under changing environmental conditions. Dai [16] proposed a robot backward + Proportional Integral Derivative (PID) force control strategy for controlling the position and attitude of the end effector, which can quickly track the desired force. Domroe [17] integrated a machine force control and feed rate control strategy to automatically adjust the trajectory of the end tool along the contours of the part surface, which is more adaptable to curved workpieces and improves the quality of grinding and rust removal.

To date, most robots still use position control to control joint angles, and robot end-trajectory error control and compensation are especially important. Accurate geometric data are usually required to effectively compensate for error [18,19,20,21]. In this study, a floating grinding actuator design is introduced, which uses only the robot’s teaching pendant to plan and run the robot, and an independent grinding controller is used to compensate for the deviation between the ideal path and the actual path to complete the grinding. The floating grinding actuator controls both the force and height of the grinding and rust removal positions. The controller is equipped with a fuzzy differential prior PID control algorithm to accurately control the grinding force, which improves the adaptive ability of the grinding actuator and the quality of grinding and rust removal.

## 2. Grinding and Rust Removal Program Design

During grinding and rust removal, the robot end effector must maintain a certain contact force and appropriate speed to perform continuous grinding and rust removal. For workpiece surfaces that are not precisely measured, it is difficult to maintain a constant grinding force for rust removal operations because of the unknown curvature, and many existing grinding robots need to premeasure the geometric surface profile, which leads to an increase in cost and a decrease in efficiency.

The path implementation concept of this study for surface grinding and rust removal is shown in Figure 1. Several way points, such as a, b, and c, were defined on the curved surface of the workpiece using a robot teaching method. Currently, most general-purpose robots can pass through a, b, and c using a linear interpolation command (usually MovL). The robot path does not match the actual curvature of the workpiece surface; however, the error is reduced if the distance between a, b, and c is reduced. More importantly, when the actuator axial height is floating, the errors due to the workpiece curvature can be compensated by controlling the actuator height using a force sensor to measure the contact force deviations and using a displacement sensor mounted on a prior position to sense the changes in curvature. It was found that when the maximum distance (Δmax) from the line between two neighboring way points to the curved surface was less than the maximum expansion (ΔL) of the actuator, which was 12 mm in the prototype design, the actuator could perform smooth and stable grinding continuously. This allows the grinding of a vehicle fender that only requires five way points along one stroke.

To evaluate the effects of grinding, the rust surfaces were processed and recognized using a specifically designed Open CV 4.5.0 image-processing software to detect the rust removal rate. The randomly captured images are converted to the HSV color space, and the HSV thresholds are set according to the color characteristics of the rust, so that the regions of rust are separated and grayscale processing is performed. The rust regions are set as pure black, the non-rust regions are set as white, and the pixel proportion of the rust region in the whole image is counted. An example of the rust-processing results is shown in Figure 2.

In the sample analysis, it was found that rust is relatively easy to clean compared with paint because of the relatively small adhesion between rust and metal surfaces. After the testing experiments, it was found that when the grinding force is controlled at 20 N, the best rust removal effect could be achieved. Based on these key data, the floating grinding actuator focused on precise grinding force regulation at 20 N.

### 2.1. Mechanisms of the Floating Grinding Actuator

The mechanical structure of the grinding actuator was enhanced from a previous design [22], as shown in Figure 3. It consists of a low-friction cylinder, two springs, a force sensor, a servo motor, a displacement sensor, and a grinding tool, which is a steel-wire brush. The cylinder was the main motion control element, the force sensor was fixed on the motor bracket through the connection plate, and the laser displacement sensor was fixed in the forward motion direction prior to the grinding tool. The servo motor output power was 500 W, rated torque was 1.6 N·m, and rated speed was 4500 rpm. The outputs of the displacement sensor and the force sensor are analog voltages with voltage values ranging from 0 to 5 V, and the measuring range of the displacement sensor was 100 ± 35 mm. The signals were converted using an A/D circuit to control the motion.

During the grinding process, when the grinding wire brush comes into contact with the workpiece, the wire brush generates a contact force on the workpiece, and the workpiece generates a reaction force of the same size on the grinding head, which is transmitted to the force sensor to measure the contact force.

### 2.2. Mathematical Modelling and Frequency Characteristics Analysis of Grinding Systems

The pneumatic system consisted of a gas source, an electrical proportional valve, and a cylinder. The pneumatic circuit is partly regulated by an electrical proportional valve to adjust the air pressure in the cylinder, which in turn controls the movement of the cylinder and causes the actuator to float up and down. Therefore, it is necessary to model the dynamics of the entire pneumatic circuit first. The wire brush grinding robot end servo mechanism and control system design originate from the dynamic modelling described in a previous study [22]. The end-grinding system transfer function is expressed as follows:(1)$$G\left(s\right)=\frac{F\left(s\right)}{U\left(s\right)}=\frac{kK_{1}R_{g}TK_{c}\left(R_{p}^{2}-r_{p}^{2}\right)\pi }{\left(Vs-kK_{2}R_{g}T\right)\left(ms^{2}+K_{v}s+2K_{d}-K_{c}\right)}$$

The relationship between the input voltage and the output grinding force of the electrical proportional valve is given by Equation (1). The hardware parameters of each part and the performance parameter table were established, and the results are shown in Table 1.

By substituting these parameters into Equation (1), the transfer function of the end-polishing system can be obtained as follows:(2)$$G\left(s\right)=\frac{14.8365}{0.0371s^{3}+32.62738s^{2}+5.855s}$$

The control system must be built on the basis of system stability. Open-loop stability reflects the system’s structural characteristics, and closed-loop stability reflects the feedback control effect, which can be used to accurately grasp the system performance boundary. An open-loop Bode diagram was obtained from the frequency domain analysis, as shown in Figure 4. The cut-off frequency is 0.66, the crossing frequency is 12.56, amplitude margin h = 50.8 (dB), phase margin γ = 15.1, h > 0 (dB) and γ > 0, so it can be judged that the system is stable.

To consider the system mapping relationship and closed-loop characteristics and to effectively analyze complex systems as well as globally judge the stability, a Nyquist diagram was used to determine the closed-loop stability. For the given transfer function, in the closed-loop state, the poles are 1.0 × 10^2^ × (−8.7927 + 0.0000i), 1.0 × 10^2^ × (−0.0009 + 0.0067i), and 1.0 × 10^2^ × (−0.0009 − 0.0067i). All poles with real parts less than 0 are in the left half-plane of the complex plane, and according to the stability determination criterion, the closed-loop system is stable, and the outputs under the bounded inputs converge to the steady state with time.

## 3. Controller Design

The control scheme of the robot grinding system is shown in Figure 5. The controller comprises two functions: grinding force control and position error compensation. The end-effector detects the grinding force in real time through the force sensor and then compares it with the preset grinding force, taking the deviation of the grinding force and the rate of change of the deviation as the inputs to the fuzzy differential prior PID algorithm. After signal processing, the control voltage u for the electrical proportional valve is generated to enable the expansion and contraction adjustment of the cylinder to realize the control of the grinding force.

Because the robot path is not initially configured to comply with the actual curvature of the work surface, the cylinder must continuously deal with the errors caused by the differences between the set path and the actual path. This can result in heavy fluctuations in the grinding force near the target force. Therefore, in this study, a compensation unit was designed to be added to the control loop. A displacement sensor mounted on the end actuator was used to detect the axial height in advance, and the readings from the displacement sensor were used to calculate the errors in the axial height, which were compensated to reduce the force fluctuations.

The distance L_1_ from the center of the cylinder end to the center of the robot’s flange during the robot grinding process is calculated using the following formula:(3)$$L_{1}=H+L+\Delta L$$
where H is the cylinder fixed height, L is the cylinder initial expansion, and ΔL is the change in the cylinder expansion.

To perform surface grinding, uphill and downhill surface situations must be considered. This study chose a cylinder with a total stroke of 50 mm, and the initial state of the cylinder was in the middle part of the total stroke, namely −25 mm≤ΔL≤+25 mm.

When the target grinding force is input into the system, the desired height is also fixed such that the initial desired height of the displacement sensor is T0 mm. The robot end-flange coordinate system is represented as {W}, the grinding end-tool coordinate system as {T}, and the fixed height from the end of the cylinder to the end of the grinding tool as L_2_. Then, there exists a translation transformation in the *Z*-axis direction for the two coordinate systems, and the corresponding matrix relation equation is as follows:(4)TwT=T(0,0,−(L1+L2))

The cylinder was changed during the grinding process, and the matrix relationship equation is as follows:(5)TwT=T(0,0,−(ΔL+L+H+L2))

In the curved workpiece grinding and rust removal process where external interference exists, the displacement sensor detects the height $T_{1}$ in advance of on each path point, calculates the expected value of the error value ΔT mm, and then compensates for the height of TwT, as follows:(6)TwT=T(0,0,−(ΔL+L+H+L2+ΔT))

Obtaining a compensated end matrix makes the contact process smoother, reduces impacts, and improves the stability of the rust removal process.

## 4. Establishment of Force Control Algorithms

Precise control of the grinding force is required to improve the quality of rust removal. In this study, a fuzzy differential prior PID control algorithm is used to precisely control the grinding force. Traditional PID control relies on fixed parameter $k_{p},k_{i},k_{d}$ and is sensitive to abrupt target value changes. The derivative term, which acts directly on error variations, easily induces oscillations. Moreover, traditional methods depend on precise models, making them unsuitable for complex surface grinding requirements. Fuzzy control does not require an accurate control model and can cope with the complex nonlinearity and uncertainty in grinding and rust removal. The differential prior PID provides differential feedback to the system output, which reduces the influence of the target grinding force change on the system. The introduction of fuzzy control into the differential prior PID algorithm is a combination of fuzzy control and PI links, with the differential term acting only on the system output. When the controlled quantity is perturbed, except for the given one, it can be restored to the stable state more quickly and maintain the system force tracking accuracy, the principle of which is shown in Figure 6.

The differential prior PID controller differentiates the output signal, does not differentiate the set value of the grinding force, and combines the differential feedback results with proportional and integral links to form a new control algorithm. The design differentiator is as follows:(7)$$\frac{u_{d}}{yt}=\frac{T_{d}s+1}{\gamma T_{d}s+1}$$
followed by(8)$$\gamma T_{d}\frac{du_{d}}{dt}+u_{d}=T_{d}\frac{dy}{dt}+y$$
where $T_{d}$ is the differentiator time constant, γ is the gain coefficient, and γ<1.

The PI link input can be derived from the PI part of the algorithm, and if T is the sampling period, the algorithm can be discretized to obtain the following:(9)$$\frac{du_{d}}{dt}=\frac{u_{d}(k)-u_{d}(k-1)}{T}$$(10)$$\frac{dy}{dt}=\frac{y(k)-y(k-1)}{T}$$(11)$$\gamma T_{d}\frac{u_{d}(k)-u_{d}(k-1)}{T}+u_{d}=T_{d}\frac{y(k)-y(k-1)}{T}+y(k)$$(12)$$u_{d}(k)=\frac{\gamma T_{d}}{\gamma T_{d}+T}u_{d}(k-1)+\frac{T_{d}+T}{\gamma T_{d}+T}y(k)-\frac{T_{d}}{\gamma T_{d}+T}y(k-1)$$

Its frequency domain expression is(13)$$u(s)=[k_{p}(1+\frac{1}{T_{i}s})E(s)+\frac{T_{d}s+1}{\gamma T_{d}s+1}y(s)]$$

From Equation (12), it can be concluded that the differential link output u_d_ (k) is only related to the previous differential link and u_d_(k − 1) system outputs y(k) and y(k − 1). The output signal of the differential link contains the value of the grinding force and the value of the speed of change of the grinding force, which is input into the proportional-integral controller as a measured value; thus, the grinding force produces a more moderate change, adapts to the impact of the grinding force, and responds to the trend of change in advance to reduce the impact of the oscillation of the grinding force on the system.

Fuzzy control is mainly composed of fuzzification, fuzzy inference, and defuzzification [23]. The set value of the grinding force, error of the feedback, and rate of change of the error are taken as inputs, and the two control parameters $k_{p}$ and $k_{i}$ of the PI controller are output to achieve the effect of adaptive tuning of the control parameters. Its control law is(14)$$u(t)=k_{p}e(t)+k_{i}(t)\int e(t)dt$$

$\Delta k_{p},\Delta k_{i}$ is the control parameter for the PI link of the fuzzy control output. The fuzzy subset is {NB, NM, NS, ZO, PS, PM, PB}, and the elements of the subset from left to right represent negative large, negative medium, negative small, zero, positive small, positive medium, and positive large, respectively. The fuzzy domain is taken as [−3, 3]; that is, the values of e and ec are in the set [−3, −2, −1, 0, 1, 2, 3]. To enhance input precision and adaptability while ensuring smooth and adjustable output, this paper selects input membership functions composed of Gaussian and triangular functions, as shown in Figure 7. The Gaussian function handles extreme input values, with σ = 0.5 controlling curve width to ensure a rapid transition to membership extremes within ±3 input range, enhancing responsiveness to large deviations. The NB (Negative Big) region uses parameters μ = −3 and σ = 0.5, while the PB (Positive Big) region uses μ = 3 and σ = 0.5. Triangular functions have vertices at (−3, −2, −1), (−2, −1, 0), (−1, 0, 1), etc. (corresponding to NM to PM regions). Output variables $\Delta k_{p}$ and $\Delta k_{i}$ adopt symmetric triangular membership distributions with vertices at [−3, −2, −1, 0, 1, 2, 3].

The fuzzy rules are determined based on the grinding force deviation, the rate of change of the grinding force deviation, and the fuzzy rules table of $\Delta k_{p}$ and $\Delta k_{i}$, as shown in Table 2. The establishment of the rule table must follow two principles: when the output quantity error is large, the error should be eliminated as soon as possible; when the output quantity error is small, the stability of the control effect should be considered. Specifically, when the actuator initially contacts the workpiece, the actual grinding force is significantly smaller than the target force. The grinding force deviation e is determined as Negative Big (NB, e = −3), and the rate of deviation change ec is also Negative Big (NB). To accelerate the response and reduce the error, the fuzzy rule table outputs $\Delta k_{p}$ = Positive Big (PB, +3) and $\Delta k_{i}$ = Negative Big (NB, −3). In uniform surface stable grinding, when e is determined to be ZO but ec is PB, it is necessary to reduce proportional gain to suppress oscillations and enhance integral action to eliminate residual errors with rule output $\Delta k_{p}$ = NM, $\Delta k_{i}$ = PM. During high-speed grinding under high-frequency interference, where sudden curvature changes cause the deviation rate to surge to PB, triggering $\Delta k_{p}$ = NM helps avoid oscillations, while $\Delta k_{i}$ = PM enhancement compensates for delays to suppress high-frequency disturbances.

The defuzzification process employs the mean of maximum (MOM) method. If multiple points in the universe of discourse of the fuzzy set attain the maximum membership degree value, the average of the horizontal coordinates of these points is taken as the representative point of the fuzzy set. Let the universe be denoted as $U=\left\{u_{1},u_{2}\ldots ,u_{n}\right\}$, where the membership function value of fuzzy set A at $u_{j}$ is $A(u_{j})$. If the membership degree at $u_{j}$ satisfies $A(u_{j})=\max[A(u)],j=1,2\ldots n$, and there are n points sharing this maximum membership value, then the representative point u is calculated as the arithmetic mean of these points:(15)$$u=\frac{1}{n}\sum_{j=1}^{n}u_{j}$$
where $u_{j}(j=1,2,\ldots ,n)$ are the points with the maximum membership degrees.

The PI control parameters $k_{p}$ and $k_{i}$ are dynamically adjusted according to fuzzy rules, as shown in the following equation:(16)$$\begin{array}{l} k_{p}=k_{p0}+\Delta k_{p}(e_{i},ec_{i},t)\\ k_{i}=k_{i0}+\Delta k_{i}(e_{i},ec_{i},t) \end{array}$$

$e_{i}$ and $ec_{i}$ are the real-time error and error variations, respectively, and $k_{p0}$ and $k_{i0}$ are the control parameters of the last calibrated PI link.

For example, when the actuator initially contacts the grinding workpiece, the actual grinding force is 15 N, significantly lower than the target force of 20 N, and the error rapidly decreases. To accelerate the response and reduce the error, the fuzzy rule table outputs $\Delta k_{p}$ = PB (+3) and $\Delta k_{i}$ = NB (−3). Since the output variables $\Delta k_{p}$ and $\Delta k_{i}$ have only a single maximum point at this moment, their values are directly taken from the universe of discourse. These are mapped to actual adjustments: $\Delta k_{p}$ as +30% and $\Delta k_{i}$ as −30%. If the previously tuned parameters $k_{p0},k_{i0}$ were 400 and 11, respectively, according to Equation (12), the final control parameters $k_{p},k_{i}$ become 520 and 7.7.

During operation, the control system completes the self-correction of the PI parameters by processing the results of the fuzzy logic rules, table look-up, and arithmetic.

To test the control effect of the fuzzy differential prior PID controller on the grinding and rust removal system, a simulation model was built, and a PID model was built for comparison. The transfer function of the control object is given by Equation (2), and the specific algorithm model is illustrated in Figure 8.

First, to verify the dynamic response speed of the algorithm, a step signal was used as the input, the contact force was set to 20 N, and the simulation time was set to 2 s. The PID parameters were manually tuned to achieve optimal parameters, with $k_{p},k_{i}$ and $k_{d}$ set to 120, 0.6, and 23, respectively. The $k_{p}$ and $k_{i}$ obtained from the fuzzy rectification are used as the PI part of the control parameters in the differential prior PID, while the system feeds back the differential parameter $k_{d}$ of 1.98, and the following results are obtained by the running model in Figure 9a.

After adding the fuzzy rectification parameters, the system starts to respond in 0.25 s and rises to the target force in 0.51 s, which is reduced by 26.92% compared to the PID regulation time, thus improving the response speed and reducing the overshoot.

To adapt to the different corrosion conditions of the workpiece, the required grinding force changes; therefore, it is necessary to consider the effect of the change in grinding force on the system. To verify the tracking effect of the grinding force, a sinusoidal signal with an amplitude of 20 was used as the input to obtain the results, as shown in Figure 9b, which shows that the fuzzy differential prior algorithm has a better force-tracking stabilization effect.

## 5. Grinding Experiments

The UR5 robot was used as the experimental mounting platform, and the robot path points and MoveL commands were used to define the grinding path and perform the surface grinding and rust removal experiments, as shown in Figure 10. The end-effector grinding control system adopts a prototype developed using an ARM-based host computer running on a Linux system(Ubuntu 20.04 LTS) and Python (3.7.12) environment. This system reduces the task scheduling latency to the microsecond level, meeting the requirements of industrial control computers for data processing and real-time control. The controller regulates the end-effector contact force by adjusting the input voltage of the electropneumatic proportional valve. The control algorithm is embedded into the controller firmware to enable autonomous force adaptation.

Grinding and rust removal experiments were carried out under the control of the PID algorithm and the fuzzy differential prior PID algorithm. The grinding and rust removal effects were tested on a bent steel plate, in which the desired grinding force was set to 20 N and the motor speed was 4000 rpm. Simultaneously, to test the force control performance of the fuzzy differential prior PID under different grinding forward speeds, grinding and rust removal experiments were conducted at grinding forward speeds of 20 mm/s, 40 mm/s, and 60 mm/s. Three round trips of grinding were performed on the curved workpieces, with five path points defined for each grinding path for grinding and rust removal. The test results of the three sets of experiments are shown in Figure 11.

The experimentally measured data are listed in Table 3 and Table 4. In the three sets of experiments, both control modes increased the fluctuation range and average error of the grinding force with an increase in the forward speed. When the forward speed is 20 mm/s, both control methods can maintain effective and stable grinding forces during testing. At a forward speed of 40 mm/s, the grinding force fluctuation of the fuzzy differential prior PID control is ±1.34, which is reduced by 37.4% compared to ±2.14 of the PID control. The average error of grinding force is reduced by 37.21%, indicating more stable force control performance. This effectively improves the control effect of robot grinding and rust removal. At a forward speed of 60 mm/s, the PID control performance deteriorates significantly, with grinding force fluctuations exceeding ±4 N. It fails to maintain stability and cannot meet the requirements of grinding and rust removal. The amount of fluctuation of the fuzzy differential prior PID control of the grinding force increases to ±2.34 N, resulting in a significant reduction in the quality of grinding and rust removal. After the analysis, it was determined that during high-speed grinding operations, the response speed of the pneumatic actuator may not match the curvature change rate of the workpiece profile, leading to delayed height compensation and subsequent contact force instability. Future design optimization could incorporate high-speed proportional valves or rigid mechanical structures to address this limitation.

As the grinding speed increases, the fuzzy differential prior PID control of the grinding force gradually oscillates; therefore, it is also necessary to check the rate of rust removal and judge the rust-removal effect. After the sample plate was rust-removed, 20 images were randomly collected using an electron microscope. Using image-processing technology, the proportion of rust accounted for was calculated, and the average rust-removal rate was obtained. The results are presented in Table 5. The rust removal rate was as high as 99.73%. The image processing process is shown in Figure 12. Simultaneously, the TR100 metal surface roughness tester was used to measure the Ra values of the grinding and rust removal areas to evaluate the surface grinding quality. For each feed speed condition, 15 measurement points were taken and averaged to reduce errors. According to the measurements shown in Table 4, when the feed rate was increased from 20 mm/s to 60 mm/s, the surface roughness Ra value under fuzzy differential prior PID control rose from 2.9 ± 0.8 μm to greater than 3.3 μm. This trend aligned with the variation in grinding force fluctuations, demonstrating that the stability of force control directly impacts surface quality.

The grinding and rust removal processes are illustrated in Figure 13a. There was a large amount of dust in the experiment, which affected the visual effect. After removing the dust attached to the surface of the workpiece, the actual effects of grinding and rust removal can be obtained, as shown in Figure 13b. The upper part is the effect of grinding and rust removal, and the lower part is the original state. It can be observed that grinding and rust removal have a high degree of finish.

In the experiment, under increasing forward speeds, the grinding force fluctuations and average error of the fuzzy differential prior PID control were significantly superior to those of the conventional PID. Especially at 40 mm/s, the fuzzy differential prior PID control reduced force fluctuations by 37.4% and maintained a rust removal rate of 97.08%, validating the robustness advantages predicted by the model. From the previous simulation results, the fuzzy differential prior PID shortened the settling time by 26.92% compared to the conventional PID in step response, with significantly reduced overshoot and better stability in sinusoidal tracking. The trends predicted by the simulation were consistent with the experimental results, indicating that the theoretical advantages of the fuzzy differential prior PID in dynamic response and stability can effectively translate into practical performance improvements. Both simulation and experimental results validated the effectiveness of the fuzzy differential prior PID algorithm.

## 6. Summary

The robotic grinding and rust removal method proposed in this study is effective for improving the quality and efficiency of rust removal. Its design features are as follows.

The end floating grinding actuator designed for robotic grinding and rust removal can sense the change in curvature of the workpiece surface through force position detection, select the optimal operating parameters according to the real-time working conditions, and adjust the controller stiffness and damping by the grinding force and axial height of the grinding position to realize continuous and constant-force grinding and rust removal of free-form surfaces.By using a floating grinding actuator and the fuzzy differential prior PID force control algorithm, self-tuning of the control parameters can be achieved, effectively reducing the jamming and jitter problems in the grinding and rust-removal process and achieving independent grinding.This paper proposes the completion of grinding and rust removal by a robot that plans the path and compensates for the error distance between the line of the path points and the curved surface contour, which can accurately track the surface contour to perform constant-force grinding and rust removal without accurately measuring the geometric contour of the workpiece.After actual testing, the rust removal rate can reach 99.73%, and good results have therefore been achieved.

This grinding actuator can be mounted on the ends of most general-purpose robots for grinding and rust removal. This method is highly effective and superior in robotic grinding and rust removal applications and provides a highly efficient, precise, and environmentally friendly solution for rust removal on large equipment in an industrial production context.

## Figures and Tables

**Figure 1 sensors-25-02204-f001:**
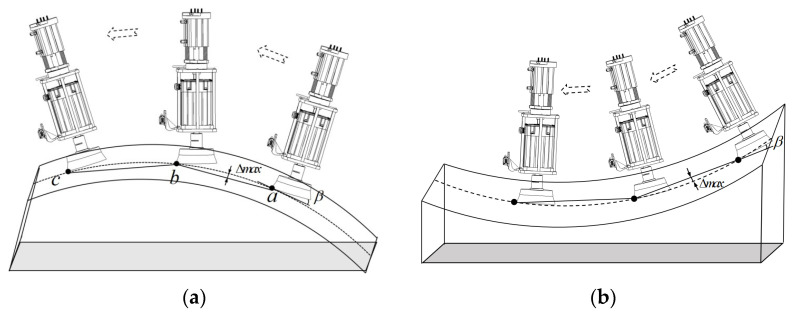
Robot grinding path determination method: (**a**) convex workpiece, (**b**) concave workpiece.

**Figure 2 sensors-25-02204-f002:**
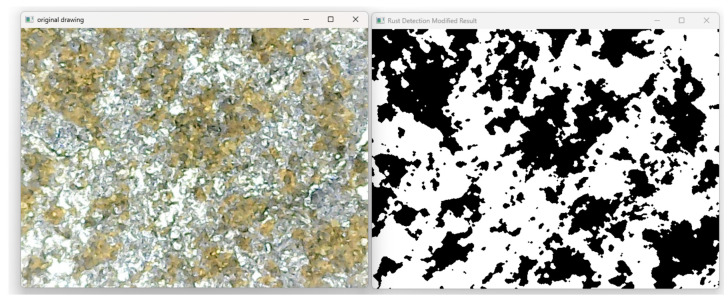
Identification of abrasive rust on a workpiece surface.

**Figure 3 sensors-25-02204-f003:**
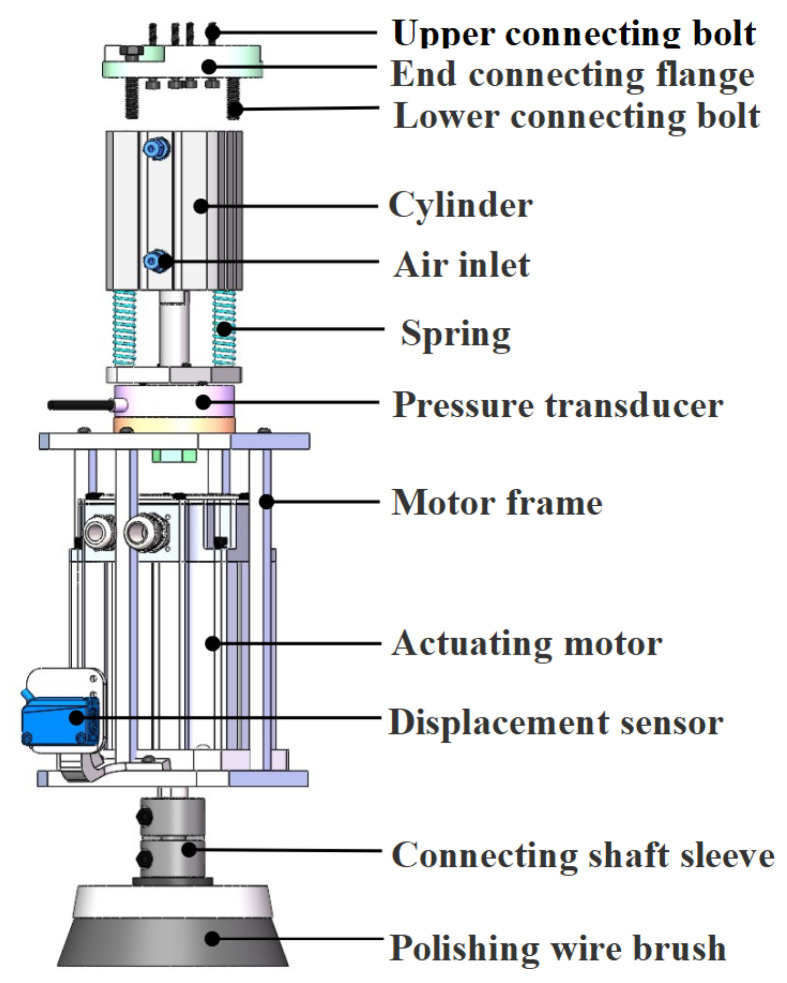
Floating grinding actuator and end grinding device force analysis diagram.

**Figure 4 sensors-25-02204-f004:**
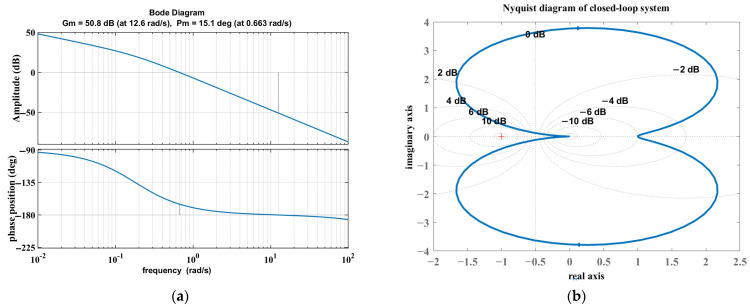
(**a**) Open-loop Bode plots; (**b**) closed-loop Nyquist plots for grinding systems.

**Figure 5 sensors-25-02204-f005:**
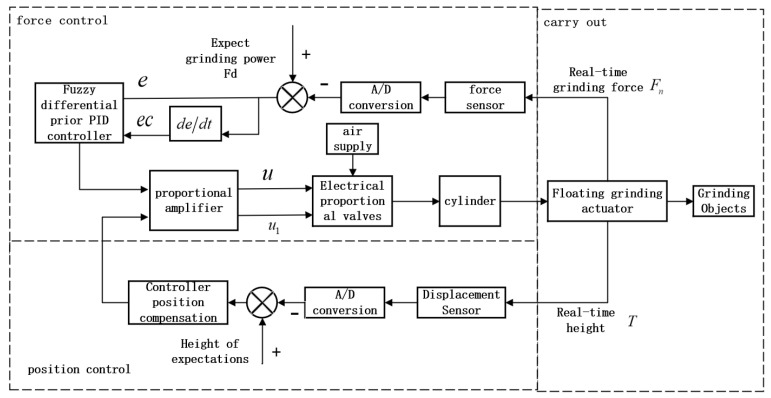
Control system block diagram.

**Figure 6 sensors-25-02204-f006:**
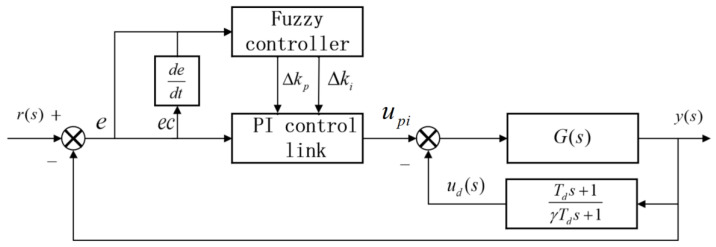
Structure of fuzzy differential prior PID controller.

**Figure 7 sensors-25-02204-f007:**
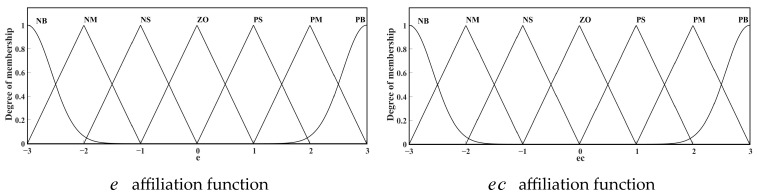
Input variable affiliation function.

**Figure 8 sensors-25-02204-f008:**
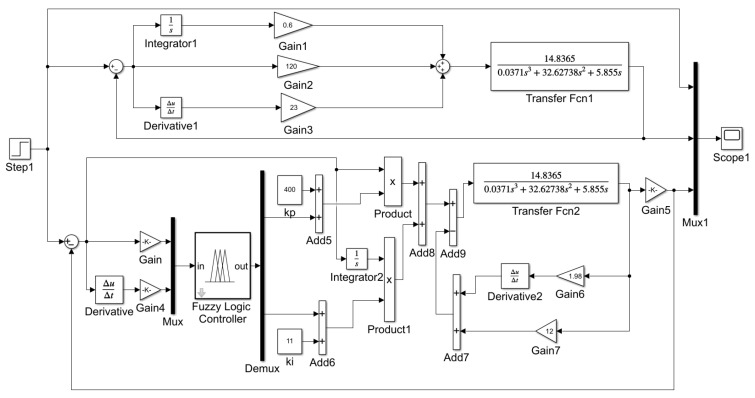
Simulation structure.

**Figure 9 sensors-25-02204-f009:**
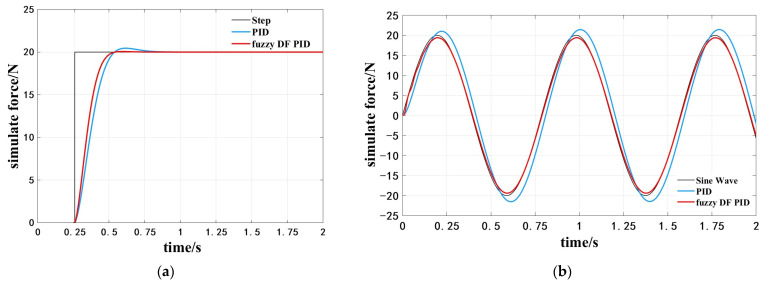
(**a**) Step response simulation; (**b**) Sinusoidal signal response simulation.

**Figure 10 sensors-25-02204-f010:**
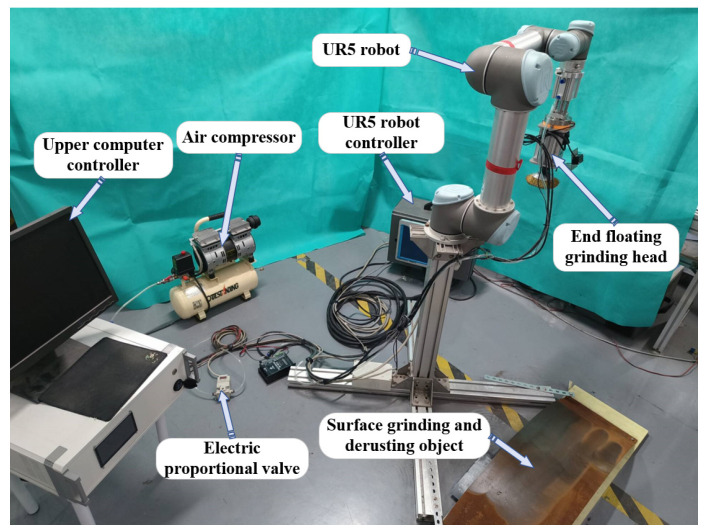
Experimental platform.

**Figure 11 sensors-25-02204-f011:**
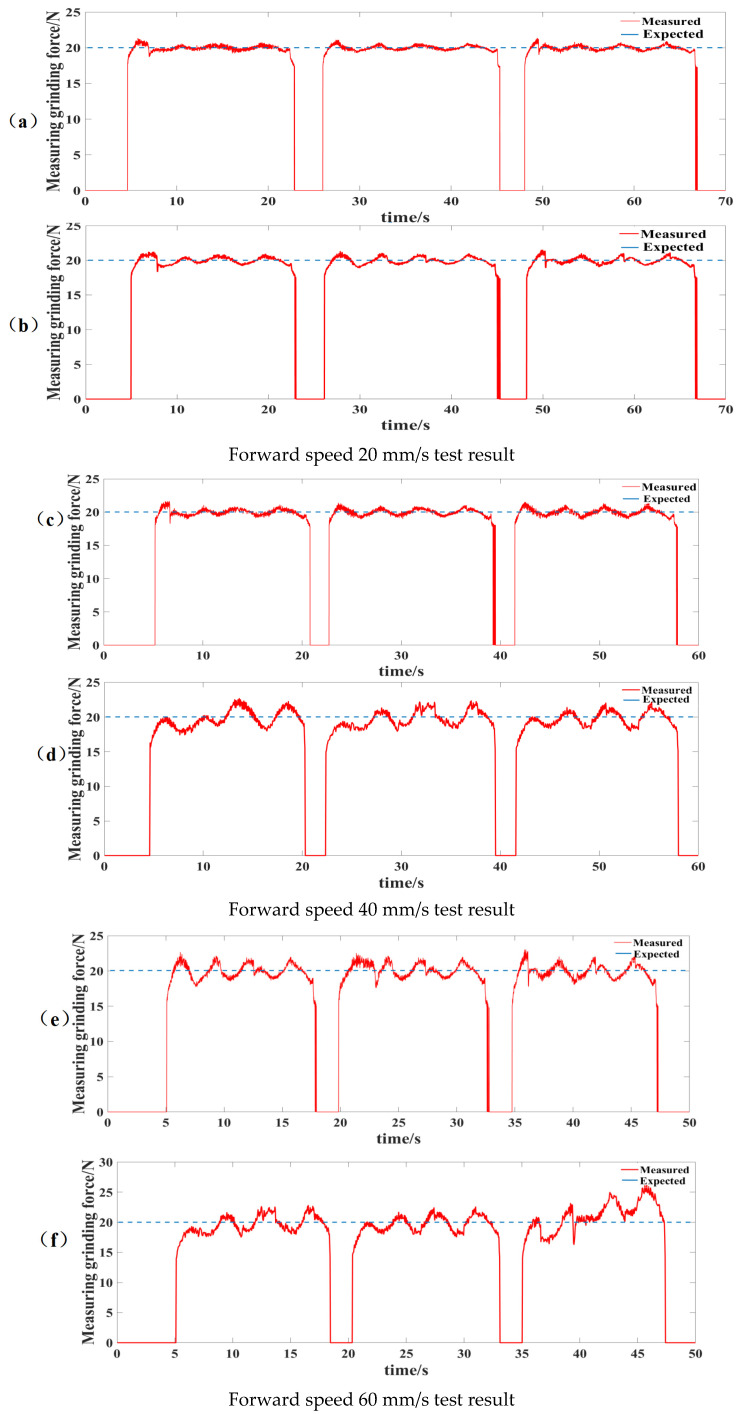
Test results at different forward speeds. (**a**) Fuzzy differential prior PID control for grinding and rust removal; (**b**) PID control for grinding and rust removal; (**c**) Fuzzy differential prior PID control for grinding and rust removal; (**d**) PID control for grinding and rust removal; (**e**) Fuzzy differential prior PID control for grinding and rust removal; (**f**) PID control for grinding and rust removal.

**Figure 12 sensors-25-02204-f012:**
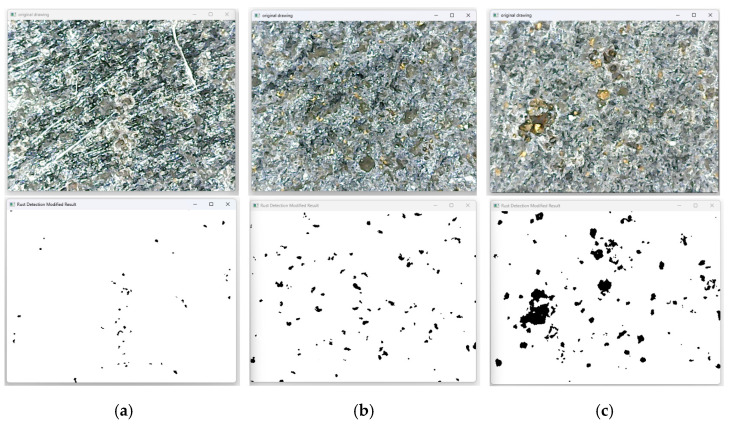
Fuzzy differential first PID control grinding rust removal rate detection results: (**a**) 20 mm/s; (**b**) 40 mm/s; (**c**) 60 mm/s.

**Figure 13 sensors-25-02204-f013:**
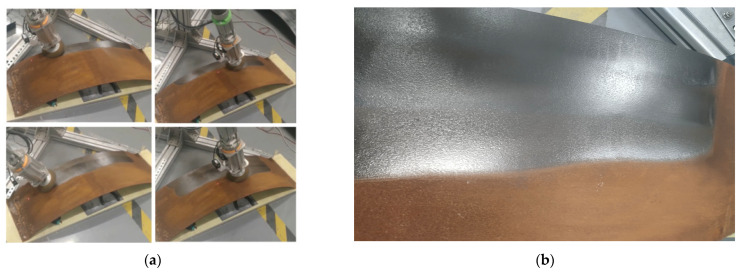
(**a**) Experimental demonstration of rust removal by grinding; (**b**) Observed effect of dust wiping after grinding.

**Table 1 sensors-25-02204-t001:** System parameters.

Parameters (Symbols)	Numerical Value (Units)
quantity (m)	3.9 kg
$\mathrm{Viscous}\;\mathrm{damping}\;\mathrm{factor}\;(K_{V}$)	0.7 N·s/m
Insulation coefficient (k)	1.4
$\mathrm{gas}\;\mathrm{constant}\;(R_{g}$)	287 N·m/kg⋅K
$\mathrm{coefficient}\;\mathrm{of}\;\mathrm{rigidity}\;(K_{V}$)	103.64 N/mm
gas temperature (T)	293 K
$\mathrm{Cylinder}\;\mathrm{cavity}\;\mathrm{radius}\;(R_{P}$)	0.025 m
$\mathrm{Coefficient}\;\mathrm{of}\;\mathrm{elasticity}\;\mathrm{of}\;\mathrm{springs}\;(K_{d}$)	51.82 N/mm
$\mathrm{Cylinder}\;\mathrm{piston}\;\mathrm{rod}\;\mathrm{radius}\;(r_{p}$)	0.01 m

**Table 2 sensors-25-02204-t002:** Table of fuzzy rules.

$\Delta k_{p},\Delta k_{i}$		EC						
		NB	NM	NS	ZO	PS	PM	PB
E	NB	PB/NB	PB/NB	PM/NM	PM/NM	PS/NS	ZO/ZO	ZO/ZO
	NM	PB/NB	PB/NB	PM/NM	PS/NM	PS/NS	ZO/ZO	NS/ZO
	NS	PM/NB	PM/NM	PM/NS	PS/NS	ZO/ZO	NS/PS	NS/PS
	ZO	PM/NM	PM/NM	PS/NS	ZO/ZO	NS/PS	NM/PM	NM/PM
	PS	PS/NM	PS/NS	ZO/ZO	NS/PS	NS/PS	NM/PM	NM/PB
	PM	PS/ZO	ZO/ZO	NS/PS	NM/PS	NM/PM	NM/PB	NB/PB
	PB	ZO/ZO	ZO/ZO	NM/PS	NM/PM	NM/PM	NB/PB	NB/PB

**Table 3 sensors-25-02204-t003:** Data table of three sets of forward speeds for fuzzy differential prior PIDs.

Forward Speed (mm/s)	Maximum Grinding Force (N)	Minimum Grinding Force (N)	Average Error of Grinding Force (N)	Amount of Grinding Force Fluctuation (N)
20	20.96	19.12	−0.13	±0.92
40	21.68	18.95	+0.27	±1.34
60	22.66	18.02	+0.31	±2.34

**Table 4 sensors-25-02204-t004:** Data table of three sets of forward speeds for PID control.

Forward Speed (mm/s)	Maximum Grinding Force (N)	Minimum Grinding Force (N)	Average Error of Grinding Force (N)	Amount of Grinding Force Fluctuation (N)
20	21.02	18.65	−0.37	±1.18
40	22.10	17.81	−0.43	±2.14
60	25.12	16.75	−0.92	±4.18

**Table 5 sensors-25-02204-t005:** Rust removal rate and Ra values corresponding to three speed conditions in fuzzy differential leading PID control experiment.

Forward Speed (mm/s)	Average Rust Removal Rate (%)	Ra Value (μm)
20	99.73	2.9 ± 0.8
40	97.08	3.0 ± 1.2
60	92.60	>3.3

## Data Availability

The data presented in this study are contained within the article.

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
